# Sarcopenia as a Risk Factor of Morbimortality in Colorectal Cancer Surgery

**Mariana Pereira [1], Ana Pereira [2], Patrícia Silva [2], Catarina Costa [3] and Sandra F. Martins [4,5,6,*]** 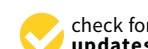

[1] School of Medicine, University of Minho, 4710-070 Braga, Portugal; mariana69055@gmail.com
[2] General Surgery Resident, Hospital de Braga, 4710-243 Braga, Portugal;
  anamaria.fppereira@gmail.com (A.P.); patriciasilva.brg@gmail.com (P.S.)
[3] Imagiology Department, Braga Hospital, 4710-243 Braga, Portugal; anakkosta@hotmail.com
[4] Coloproctology Unit, Department of General Surgery, Hospital de Braga, 4710-243 Braga, Portugal
[5] Life and Health Science Research Institute (ICVS), School of Medicine, University of Minho,
  4710-070 Braga, Portugal
[6] Life and Health Sciences Research Institute (ICVS)/3B's-PT Government Associate Laboratory,
  Braga/Guimarães, 4710-070 Braga, Portugal
*  Correspondence: sandramartins@med.uminho.pt; Tel.: +351-253604827; Fax: +351-253604847

**Abstract:** Background: Colorectal cancer (CRC) surgery is associated with high rates of postoperative morbimortality. Sarcopenia has been identified as an independent predictor of these surgical outcomes. Methods: A sample of 272 patients who underwent CRC surgery between January 2005 and May 2010 at Braga Hospital, was selected. Sarcopenia was defined by the skeletal muscle mass index, measured by preoperative computed tomography (CT), at L3 level, using ImageJ®software. Associations between sarcopenia and qualitative variables were analyzed by Chi-Square Test ($\chi^2$) or Fisher's Exact Test and, for quantitative variables, by Mann-Whitney Test. A multivariate logistic regression was performed to assess if sarcopenia was an independent predictor of major morbidity. The overall and recurrence-free survivals were analyzed by Kaplan-Meier method and multivariate Cox regression was performed for recurrence-free survival. Results: The prevalence of sarcopenia was 19.1%. Sarcopenia was associated with male gender, no CRC family history and colon tumour ($p < 0.001$, $p = 0.029$ and $p = 0.017$, respectively). The presence of sarcopenia was associated with postoperative morbidity Clavien–Dindo classification ($p = 0.003$), and sarcopenia was an independent predictor for major complications (grade ≥ III) ($p = 0.003$). Conclusions: The evaluation of sarcopenia in patients undergoing CRC surgical resection allows to predict a higher probability of major postoperative morbimortality.

**Keywords:** sarcopenia; colorectal cancer; postoperative morbimortality

---

## 1. Introduction

Colorectal cancer (CRC) is the third most common cancer worldwide, with an estimate of 1.4 million new cases per year. It is the second leading cause of death related to cancer, being responsible for 693,900 deaths per year, according to GLOBOCAN 2012 [1–3].

Sarcopenia is a geriatric syndrome characterised by progressive loss of skeletal muscle mass and function (strength and/or physical performance), with a higher risk of physical disability, poor quality of life and death [4,5].

The loss of muscle mass can be assessed by computed tomography (CT), magnetic resonance imaging (MRI), dual energy x-ray absorptiometry (DXA) or bioelectrical impedance analysis. The decrease of muscle strength can be measured by handgrip strength or knee extension/flexion,

and the decrease in physical performance by the short physical performance battery test, the timed get-up-and-go test, gait speed or the stair climb power test [4]. There are no consensual cut-off points for sarcopenia definition in the literature and they also depend upon the measurement technique chosen. Nevertheless, the European Working Group on Sarcopenia in Elderly People (EWGSOP) recommends as cut-off points two standard deviations below the mean reference value [4].

With a growing interest in evaluating the influence of sarcopenia in cancer patients, as CRC patients, different diagnostic techniques have been studied, and MRI, CT and DXA were defined as the gold standard in these patients. As CT is generally used for clinical tumour staging, it becomes an accessible method for the diagnosis of sarcopenia [6–8]. Through a single abdominal cross-sectional CT image, at the third lumbar vertebrae (L3), it is possible to estimate the total body mass by evaluating the skeletal muscle index $(cm^2/m^2)$, which is calculated by the sum of skeletal muscle areas, at L3 level, and normalised for stature [9–12]. Based on this diagnostic method, the prevalence of sarcopenia varied from 15% to 71% in CRC patients [9–12].

In these patients, an association between sarcopenia and negative outcomes was found, i.e., increased global and cancer-related mortality, higher recurrence of disease, higher postoperative hospital stays, higher risk of infection and other postoperative complications, as well as the need for rehabilitation care and superior toxicity from chemotherapy [12–16].

Thus, with the rise of elderly population, the number of patients with CRC at this age group is growing, which increases the risk of perioperative complications, with higher rates of morbimortality and therapeutic failure. Therefore, we aimed to evaluate the influence of sarcopenia in these surgical outcomes, as an independent predictor of a functional and morbimortality compromise.

## 2. Results

### 2.1. Relationship between Sarcopenia and the Clinical Pathological Data

Among the 272 patients, 52 (19.1%) had sarcopenia and 220 (81.9%) did not; 167 (61.4%) were males and 105 (38.6%) females, with ages between 31 and 92 years (Mdn = 72.0, IQR = 17.0). Table 1 shows the descriptive statistics of the clinical and pathological data, as well as their relationship with sarcopenia.

**Table 1.** Relationship between sarcopenia and clinicopathological factors.

| | All Patients (*n* = 272) | Sarcopenic (*n* = 52) | Non-Sarcopenic (*n* = 220) | Statistics Test |
|---|---|---|---|---|
| **Age**, *Mdn (IQR)* | 72.0 (17.0) | 71.0 (18.0) | 72.0 (16.0) | $U = 5460, p = 0.609,$ $r = -0.031$ |
| **Gender**, *n* (%) | | | | |
| Male | 167 (61.4) | 46 (88.5) | 121 (55.0) | $\chi^2 (1) = 19.9, p < 0.001$ *, |
| Female | 105 (38.6) | 6 (11.5) | 99 (45.0) | $\Phi = -0.27$ |
| **CRC family history** [a], *n* (%) | | | | |
| Absent | 237 (92.6) | 48 (100.0) | 189 (90.9) | Fisher's test, $p = 0.029$ *, |
| Present | 19 (7.40) | 0 (0.00) | 19 (9.10) | $\Phi = -0.14$ |
| **Clinical presentation**, *n* (%) | | | | |
| Asymptomatic | 48 (17.6) | 6 (11.5) | 42 (19.1) | $\chi^2 (1) = 1.65, p = 0.230,$ |
| Symptomatic | 224 (82.4) | 46 (88.5) | 178 (80.9) | $\Phi = 0.078$ |
| **Tumour site**, *n* (%) | | | | |
| Colon | 193 (71.9) | 44 (84.6) | 149 (67.7) | $\chi^2 (1) = 5.82, p = 0.017,$ |
| Rectum | 79 (29.0) | 8 (15.4) | 71 (32.3) | $\Phi = -0.15$ |
| **Macroscopic aspect** [a], *n* (%) | | | | |
| Polypoid | 135 (53.6) | 24 (52.2) | 111 (53.9) | |
| Ulcerative | 66 (26.2) | 9 (19.6) | 57 (27.7) | |
| Infiltrative | 24 (9.50) | 3 (6.50) | 21 (10.2) | Fisher's test, $p = 0.085,$ |
| Exophytic | 26 (10.30) | 10 (21.7) | 16 (7.80) | $\Phi_c = 0.19$ |
| Villous | 1 (0.40) | 0 (0.00) | 1 (0.50) | |

**Table 1.** *Cont.*

| | All Patients (*n* = 272) | Sarcopenic (*n* = 52) | Non-Sarcopenic (*n* = 220) | Statistics Test |
|---|---|---|---|---|
| **Measurement** [a], *n* (%) | | | | |
| ≤45 mm | 167 (64.5) | 27 (56.3) | 140 (66.4) | $\chi^2$ (1) = 1.74, *p* = 0.242, |
| >45 mm | 92 (35.5) | 21 (43.8) | 71 (33.6) | Φ = 0.082 |
| **Histological type**, *n* (%) | | | | |
| Adenocarcinoma | 245 (90.1) | 48 (92.3) | 197 (89.5) | Fisher's test, *p* = 0.834, |
| Mucinous | 26 (9.60) | 4 (7.70) | 22 (10.0) | $\Phi_c$ = 0.043 |
| Signet ring cells | 1 (0.40) | 0 (0.00) | 1 (0.50) | |
| **Differentiation** [a], *n* (%) | | | | |
| Well differentiated | 125 (47.9) | 22 (45.8) | 103 (48.4) | |
| Moderately differentiated | 102 (39.1) | 20 (41.7) | 82 (38.5) | Fisher's test, *p* = 0.959, |
| Poorly differentiated | 33 (12.6) | 6 (12.5) | 27 (12.7) | $\Phi_c$ = 0.038 |
| Undifferentiated | 1 (0.40) | 0 (0.00) | 1 (0.50) | |
| **Venous invasion** [a], *n* (%) | | | | |
| Absent | 146 (57.5) | 28 (59.6) | 118 (57.0) | $\chi^2$ (1) = 0.009, *p* = 0.870, |
| Present | 108 (42.5) | 19 (40.4) | 89 (43.0) | Φ = -0.020 |
| **Lymphatic invasion** [a], *n* (%) | | | | |
| Absent | 103 (41.0) | 19 (40.4) | 84 (41.2) | $\chi^2$ (1) = 0.01, *p* = 1.000, |
| Present | 148 (59.0) | 28 (59.6) | 120 (58.8) | Φ = 0.006 |
| **Stage** [a], *n* (%) | | | | |
| I | 43 (16.2) | 7 (14.0) | 36 (16.7) | |
| II | 87 (32.7) | 17 (34.0) | 70 (32.4) | $\chi^2$ (3) = 0.93, *p* = 0.817, |
| III | 93 (35.0) | 16 (32.0) | 77 (35.6) | $\Phi_c$ = 0.059 |
| IV | 43 (16.2) | 10 (20.0) | 33 (15.3) | |
| ***CEA*** [a], *n* (%) | | | | |
| ≤10 ng/mL | 197 (82.4) | 35 (77.8) | 162 (83.5) | $\chi^2$ (1) = 0.83, *p* = 0.386, |
| >10 ng/mL | 42 (17.6) | 10 (22.2) | 32 (16.5) | Φ = 0.059 |

[a] Does not reach the total n due to the existence of missing values. $\chi^2$—Chi-square test; *U*—Mann–Whitney test; *n*—absolute frequency; %—relative frequency; *Mdn*—median; *IQR*—interquartile range; Φ—Phi; $\Phi_c$—Cramér's V; *p*—level of significance.

Gender was significantly associated with sarcopenia (*p* < 0.001) and most patients were male in both groups, with (88.5%) or without sarcopenia (55.0%). A statistically significant association between sarcopenia and CRC family history (*p* = 0.029) was found. Patients with sarcopenia had no family history of CRC (100.0%), as well as most of the patients without sarcopenia (90.9%). There was a significant association between sarcopenia and tumour site (*p* = 0.017), and most patients with (84.6%) and without sarcopenia (67.7%) had a colon tumour.

Total abdominal muscle areas were significantly different between patients with or without sarcopenia (*p* < 0.001), being lower in sarcopenic patients. Likewise, significant differences were found in the skeletal muscle index (SMI) (*p* < 0.001), with lower values in sarcopenic patients too (Table 2).

**Table 2.** Relationship between sarcopenia and body composition parameters.

| | All Patients | Sarcopenic | Non-Sarcopenic | Statistics Test |
|---|---|---|---|---|
| **Total abdominal muscle area, L3 (cm$^2$)**, *Mdn (IQR)* | 142 (47.0) | 131 (24.9) | 149 (51.2) | *U* = 3525, *p* < 0.001 *, *r* = −0.26 |
| **Skeletal muscle index (*SMI*) (cm$^2$/m$^2$)**, *Mdn (IQR)* | 53.7 (12.5) | 48.2 (8.50) | 56.2 (12.6) | *U* = 1955, *p* < 0.001 *, *r* = −0.45 |

*U*—Mann–Whitney test; *Mdn*—median; *IQR*—interquartile range; *p*—level of significance.

## 2.2. Relation between Sarcopenia and Surgical Outcomes

The associations between postoperative morbidity and sarcopenia are shown in Table 3, in which a significant association between sarcopenia and the Clavien–Dindo Classification was found (*p* = 0.003). Most of the sarcopenic patients had a CDC grade IIIb (41.2%), while most of the non-sarcopenic patients had CDC grades I (33.3%) and II (34.7%). A significant association between sarcopenia and

CDC grade ≥ III was found (*p* = 0.002), with most of the sarcopenic patients with grade ≥ III (76.5%), whereas most of the non-sarcopenic patients presented a grade < III (66.7%).

**Table 3.** Relationship between sarcopenia and postoperative morbidity.

| | All Patients (*n* = 272) | Sarcopenic (*n* = 52) | Non-Sarcopenic (*n* = 220) | Statistics Test |
|---|---|---|---|---|
| **Morbimortality** [a], *n* (%) | | | | |
| Absent | 176 (65.3) | 34 (66.7) | 141 (65.0) | $\chi^2$ (1) = 0.052, *p* = 0.871, |
| Present | 92 (34.7) | 17 (33.3) | 76 (35.0) | Φ = −0.014 |
| **Surgical site infection** [a], *n* (%) | | | | |
| Absent | 235 (87.7) | 45 (88.2) | 190 (87.6) | $\chi^2$ (1) = 0.018, *p* = 1.000, |
| Present | 33 (12.3) | 6 (11.8) | 27 (12.4) | Φ = −0.008 |
| **Anastomosis leakage**[a], *n* (%) | | | | |
| Absent | 246 (91.8) | 45 (88.2) | 201 (92.6) | Fisher's test, *p* = 0.392, |
| Present | 22 (8.20) | 6 (11.8) | 16 (7.40) | Φ = 0.063 |
| **Intraabdominal abscess** [a], *n* (%) | | | | |
| Absent | 255 (95.1) | 48 (94.1) | 207 (95.4) | Fisher's test, *p* = 0.718, |
| Present | 13 (4.90) | 3 (5.90) | 10 (4.60) | Φ = 0.023 |
| **Other infections** [a,b], *n* (%) | | | | |
| Absent | 261 (97.4) | 51 (100.0) | 210 (96.8) | Fisher's test, *p* = 0.353, |
| Present | 7 (2.60) | 0 (0.00) | 7 (3.20) | Φ = −0.079 |
| **Pulmonary complications** [a], *n* (%) | | | | |
| Absent | 254 (94.8) | 48 (94.1) | 206 (94.9) | Fisher's test, *p* = 0.734, |
| Present | 14 (5.20) | 3 (5.90) | 11 (5.10) | Φ = 0.014 |
| **Other complications** [a,c], *n* (%) | | | | |
| Absent | 248 (92.5) | 48 (94.1) | 200 (92.2) | Fisher's test, *p* = 0.774, |
| Present | 20 (7.50) | 3 (5.90) | 17 (7.80) | Φ = −0.029 |
| **Clavien-Dindo classification**, *n* (%) | | | | |
| Grade I | 27 (29.3) | 2 (11.8) | 25 (33.3) | |
| Grade II | 28 (30.4) | 2 (11.8) | 26 (34.7) | |
| Grade IIIb | 21 (22.8) | 7 (41.2) | 14 (18.7) | Fisher's test, *p* = 0.003 *, |
| Grade Iva | 1 (1.10) | 1 (5.90) | 0 (0.00) | $\Phi_c$ = 0.44 |
| Grade IVb | 1 (1.10) | 1 (5.90) | 0 (0.00) | |
| Grade V | 14 (15.2) | 4 (23.5) | 10 (13.3) | |
| **≥ Grade II**, *n* (%) | | | | |
| Absent | 26 (28.3) | 2 (11.8) | 24 (32.0) | Fisher's test, *p* = 0.137, |
| Present | 66 (71.7) | 15 (88.2) | 51 (68.0) | Φ = 0.17 |
| **≥ Grade III**, *n* (%) | | | | |
| Absent | 54 (58.7) | 4 (23.5) | 50 (66.7) | $\chi^2$ (1) = 10.6, *p* = 0.002 *, |
| Present | 38 (41.3) | 13 (76.5) | 25 (33.3) | Φ = 0.34 |

[a] Does not reach the total n due to the existence of missing values. [b] Urinary tract infection, pneumonia, endocarditis and cholecystitis. [c] Hemorrhage, anemia, pneumoperitoneum, paresthesia, hypertensive crisis, fever, seroma of the surgical wound, hematoma of the surgical wound, intestinal adhesions and urinary retention. $\chi^2$—Chi-square test; *n*—absolute frequency; %—relative frequency; Φ—Phi; $\Phi_c$—Cramér's V; *p*—level of significance.

Univariate and multivariate logistic regression analyses were performed to assess whether sarcopenia was an independent predictor of postoperative complications with grade ≥ III (Table 4).

In univariate logistic regression analyses, sarcopenia, anastomosis leakage and surgical site infection were significant predictors. Intraabdominal abscess and pulmonary complications showed a trend close to significance. Thus, these predictors were included in a multivariate logistic regression model, and age and gender were also included as control data. This model was significant (*p* < 0.001; $R^2$ Nagelkerke = 0.57; percentage of correctly predicted cases = 82.6%), and having sarcopenia was found to be associated with a higher probability of major complications (*p* = 0.003), as well as the presence of anastomosis leakage (*p* < 0.001), or intraabdominal abscess (*p* = 0.031).

There were no significant associations between sarcopenia and other surgical outcomes (Table 5).

*2.3. Relation between Sarcopenia and Overall Survival*

A total of 159 (58.5%) deaths were recorded, 34 (65.4%) in sarcopenic patients and 125 (57.1%) in non-sarcopenic patients. The median of overall survival was lower in sarcopenic group (*Mdn* = 38.0,

*IQR* = 95.0), compared with non-sarcopenic group (*Mdn* = 87.0, *IQR* = 87.0), although, sarcopenia did not significantly correlate with overall survival on Kaplan–Meier method and the Log-rank test (*p* = 0.160) (Figure 1).

**Table 4.** Univariate and multivariate logistic regression analyses of the predictors for major postoperative complications (Clavien–Dindo classification ≥ grade III).

|  | Univariate Analysis | | Multivariate Analysis * | |
|---|---|---|---|---|
|  | Unadjusted *OR* (*CI* 95%) | *p* Value | Adjusted *OR* (*CI* 95%) | *p* Value |
| **Age** | 1.02 (0.98, 1.07) | 0.328 | 1.04 (0.98, 1.11) | 0.155 |
| **Gender** (male vs. female) | 0.66 (0.26, 1.64) | 0.369 | 1.30 (0.35, 4.90) | 0.685 |
| **Tumour site** (colon vs. rectum) | 0.85 (0.36, 1.99) | 0.706 | | |
| **Stage** (I, II vs. III, IV) | 0.87 (0.38, 2.01) | 0.747 | | |
| **Total abdominal muscle area** | 0.99 (0.98, 1.01) | 0.272 | | |
| **Sarcopenia** (absent vs. present) | 6.50 (1.92, 22.0) | 0.003 | 13.6 (2.42, 76.7) | 0.003 * |
| **Anastomosis leakage** (absent vs. present) | 28.9 (6.14, 136) | <0.001 | 42.9 (6.75, 272) | <0.001 * |
| **Intraabdominal abscess** (absent vs. present) | 2.61 (0.78, 8.73) | 0.119 | 5.42 (1.16, 25.3) | 0.031 * |
| **Surgical site infection** (absent vs. present) | 0.39 (0.16, 0.97) | 0.044 | 0.76 (0.19, 3.04) | 0.700 |
| **Other infections** (absent vs. present) | 0.27 (0.030, 2.36) | 0.234 | | |
| **Pulmonary complications** (absent vs. present) | 0.34 (0.087, 1.30) | 0.113 | 0.44 (0.071, 2.69) | 0.371 |

*OR*—Odds ratio; *CI*—confidence interval; *p*—level of significance; vs.—versus. * $R^2$ = 0.42 (Cox & Snell), 0.57 (Nagelkerke). Model $\chi^2$ (7) = 50.64, *p* < 0.001.

**Table 5.** Relationship between sarcopenia and other postoperative morbimortality outcomes.

|  | All Patients (*n* = 272) | Sarcopenic (*n* = 52) | Non-Sarcopenic (*n* = 220) | Statistics Test |
|---|---|---|---|---|
| **Postoperative hospital stay (days)**, *Mdn (IQR)* | 7.00 (4.00) | 7.00 (4.00) | 7.00 (4.00) | *U* = 5589, *p* = 0.915, *r* = 0.007 |
| **Readmission**, *n* (%) | | | | |
| Absent | 265 (97.4) | 50 (96.2) | 215 (97.7) | Fisher's test, *p* = 0.622, Φ = 0.039 |
| Present | 7 (2.60) | 2 (3.80) | 5 (2.30) | |
| **30-day mortality** [a], *n* (%) | | | | |
| Absent | 251 (92.6) | 46 (88.5) | 205 (93.6) | Fisher's test, *p* = 0.235, Φ = 0.078 |
| Present | 20 (7.40) | 6 (11.5) | 14 (6.40) | |

[a] *Does not reach the total n due to the existence of missing values. U—Mann–Whitney test; n—absolute frequency; %—relative frequency; Mdn—median; IQR—interquartile range; Φ—Phi; p—level of significance.*

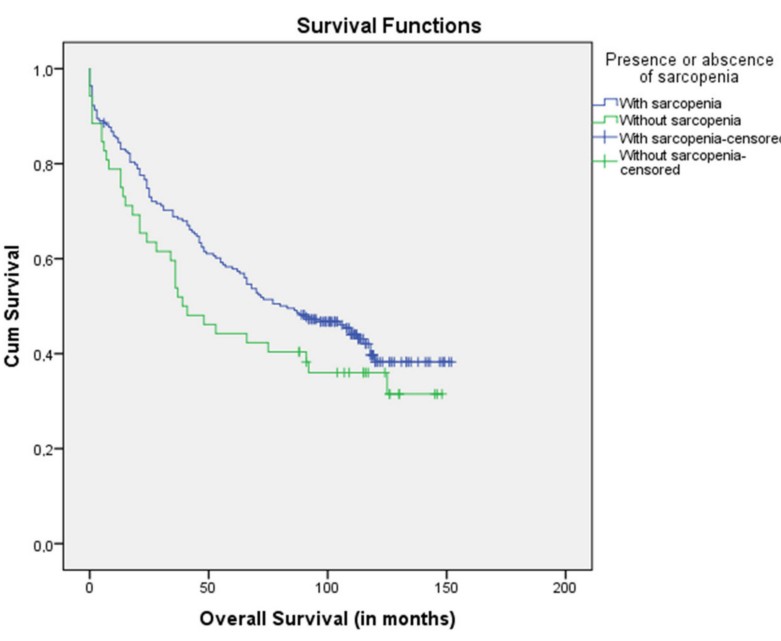

**Figure 1.** Overall survival in sarcopenic and non-sarcopenic patients (in months). *p* = 0.160.

## 2.4. Relationship between Sarcopenia and Disease-Free Survival

A total of 76 (29.6%) recurrences of the disease were listed, 19 (38%) in the group with sarcopenia and 57 (27.5%) in the group without sarcopenia. The median of recurrence-free survival was lower in the sarcopenic group (*Mdn* = 26.0, *IQR* = 77.0), compared with the non-sarcopenic group (*Mdn* = 68.0, *IQR* = 97.0). On the Kaplan–Meier method and Log-rank test (Figure 2), results came close to statistically significant values (*p* = 0.055), so, a Cox regression was performed.

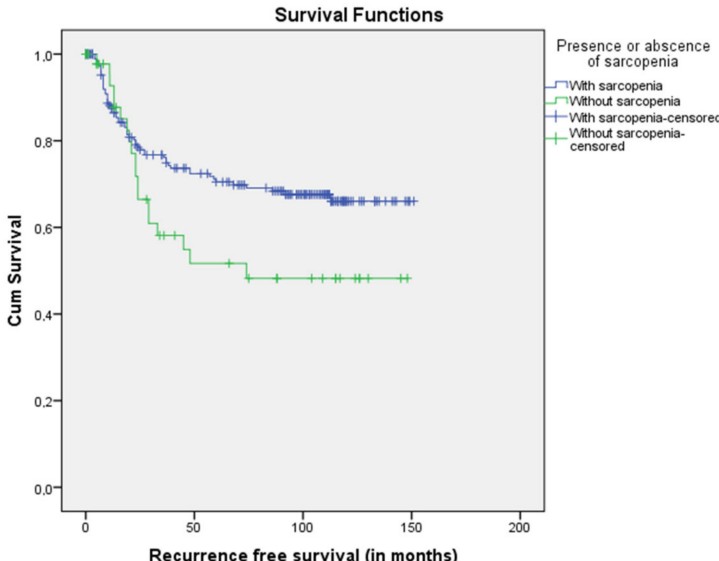

**Figure 2.** Recurrence-free survival in sarcopenic and non-sarcopenic patients (in months). *p* = 0.055.

Univariate and multivariate Cox regression analyses were performed to assess whether sarcopenia was an independent predictor of recurrence-free survival (Table 6).

**Table 6.** Univariate and multivariate Cox regression analyses of the predictors for recurrence-free survival.

| | Univariate Analysis | | Multivariate Analysis | |
|---|---|---|---|---|
| | **Unadjusted *HR* (*CI* 95%)** | ***p* Value** | **Adjusted *HR* (*CI* 95%)** | ***p*** |
| **Age** | 1.00 (0.98, 1.02) | 0.941 | 1.02 (1.00. 1.05) | 0.118 |
| **Gender** (male vs. female) | 0.94 (0.59, 1.52) | 0.813 | 1.11 (0.61. 2.02) | 0.733 |
| **CRC family history** (absent vs. present) | 0.48 (0.15, 1.52) | 0.211 | | |
| **Tumour site** (colon vs. rectum) | 1.03 (0.63, 1.70) | 0.899 | | |
| **Stage** (I, II vs. III, IV) | 2.33 (1.45, 3.24) | <0.001 | 1.21 (0.60. 2.45) | 0.599 |
| **Total abdominal muscle area** | 1.00 (1.00, 1.00) | 0.357 | | |
| **Sarcopenia** (absent vs. present) | 1.65 (0.98, 2.78) | 0.058 | 1.83 (0.96. 3.49) | 0.066 |
| **Differentiation** (G1, G2 vs. G3, G4) | 1.53 (0.78, 2.97) | 0.215 | | |
| **Venous invasion** (absent vs. present) | 2.81 (1.75, 4.50) | <0.001 | 3.05 (1.64. 5.66) | <0.001 * |
| **Lymphatic invasion** (absent vs. present) | 1.69 (1.03, 2.77) | 0.036 | 0.95 (0.46. 1.97) | 0.887 |
| **CEA** (≤10 versus >10 ng/mL) | 2.86 (1.63, 5.03) | <0.001 | 3.21 (1.69, 6.09) | <0.001 * |

*HR*—Hazard ratio; *CI*—confidence interval; *p*—level of significance; G1—well differentiated; G2—moderately differentiated; G3—poorly differentiated; G4—undifferentiated. vs.—versus.

In univariate Cox regression analyses, stage III and IV, vascular invasion, lymphatic invasion and CEA > 10 ng/mL were statistically significant predictors of recurrence-free survival. Sarcopenia and tumour differentiation showed a trend close to significance. Thus, these predictors were included in the multivariate Cox regression model, and age and gender were also included as control variables. This model was significant (*p* <0.001), however, sarcopenia did not show to be a significant predictor.

The presence of vascular invasion and CEA values > 10 ng/mL were significant predictors of lower recurrence-free survival ($p$ = 0.002 and $p$ = 0.017, respectively).

## 3. Discussion

CRC is associated with elderly age and, as curative therapy goes through surgical resection, it is associated with higher rates of postoperative morbidity and mortality. Thus, it is essential to evaluate the surgical risk of patients undergoing this treatment, in order to select patients in whom the benefits overcome the risks, and also those who are more likely to benefit from a tight postoperative follow up [17,18]. Therefore, this study sought to evaluate the influence of sarcopenia, as an independent predictor of negative surgical outcomes.

The prevalence of sarcopenia (19.1%) was lower than the prevalence reported by Nakanishi et al. (60.0%) and Lieffers et al. (38.8%). This could be explained by the different methods of diagnosis and cut-off values. These studies also used preoperative abdominal CT for the assessment of SMI, as well as the same cut-off values (Prado et al.) [13,19]. Nevertheless, in the systematic review of Malietzis et al., the prevalence of sarcopenia varied from 15.9% to 71% in CRC patients, which can be explained by the heterogeneity between samples [12].

With regard to sociodemographic data, there was a significant association between sarcopenia and gender, with a higher prevalence in males. In the literature, the results are controversial and do not always report this association. Nevertheless, in general, a higher prevalence of sarcopenia in males is known [13,19]. No significant differences were found between sarcopenia and age. The same results were found by Nakanishi et al. [19]. However, controversial results about sociodemographic data are found in the literature, and other studies report a significant association between sarcopenia and older ages [13,20]. Since this study reported a median of 72 years in the total sample, as well as in the non-sarcopenic group, and a median of 71 years in the sarcopenic group, it reflects an elderly population with CRC, which may explain the absence of significant differences.

With regard to surgical outcomes, sarcopenia did not show a significant association with postoperative complications; however, it was found that sarcopenia is an independent predictor of major postoperative complications (grade ≥ III). Likewise, Nakanishi et al. found a significant association with postoperative morbidity of all grades, grade ≥ III and especially degree ≥ II, and it was later verified that sarcopenia was an independent predictor of grade ≥ II postoperative complications [19]. These results were also reported by Jones et al. and other studies, although some used different methods of sarcopenia diagnosis [12,14].

There was no significant association between sarcopenia and the type of complication (anastomosis leakage, intraabdominal abscess, surgical wound infection and other infections). These results are also reported by Nakanishi et al. and Reisinger et al., except for the presence of other infections or anastomosis leakage, respectively [19,21]. Lieffers et al. reported that sarcopenia was an independent predictor of the presence of infections, however, they did not analyze the details of infections and other complications [13].

There was not shown any association between length of hospital stay and sarcopenia, with a median of 7.0 days in sarcopenic patients, compared to 7.0 days in non-sarcopenic patients. However, if other confounding data were controlled, such as the fact that it was an elective or emergency surgery, the results could change. These results were also reported by Jones et al. (6.23 vs. 7.69 days) [14]. On the other hand, Lieffers et al. and Nakanishi et al. found that sarcopenia was a predictor of longer postoperative hospital stays, 15.9 versus 12.3 days, and 16.3 versus 19.4 days, respectively [13,19].

Although the medians of overall and recurrence-free survivals in sarcopenic patients were shorter than in non-sarcopenic patients, they were not correlated with sarcopenia. Nakanishi et al. found equal results [19]. Black et al. found an association between sarcopenia and shorter overall survival, on Kaplan Meier method, but the results were no longer significant when analyzed in a multivariate Cox regression [20].

There was no significant association between 30-day postoperative mortality and sarcopenia. In contrast, Reisinger et al. found a significant association with the presence of sarcopenia [21]. Thus, our results are not consistent with the literature, although the frequency of this outcome in patients with sarcopenia (11.5%) was almost double compared with patients without sarcopenia (6.4%). On the other hand, it has been described that the presence of negative postoperative outcomes is more important than pre-surgical risk factors in survival after a major surgery. So, the prevention of these outcomes, with an improvement of hospitalization care and integration on specific programs that aimed this purpose, may be essential to improve the survival of these patients [18].

Some limitations can be identified in the present study, the main one being the fact that this is a retrospective study, so data collection is limited by the information present in the clinical process, which may influence the interpretation of the results. Although most studies evaluate sarcopenia only based on the loss of muscle mass, its definition comprehends a state of progressive loss of muscle mass and function, which, along with the lack of consensus in the diagnostic method and cut-off values, are limitations of the study. Thus, new studies with larger samples are necessary in order to determine specific cut-offs for each population and, therefore, to overcome some differences found between studies.

## 4. Materials and Methods

### 4.1. Patients

We retrospectively analyzed 512 patients with colorectal adenocarcinoma who underwent surgical resection at Hospital de Braga from January 2005 to May 2010. A sample of 272 patients was selected, based on inclusion and exclusion criteria. Patients who underwent neoadjuvant chemotherapy and/or radiotherapy (*n* = 15), without preoperative abdominal CT available on digital storage, up to 6 months before surgery (*n* = 205), without a visualization of the total abdominal muscle area (n = 18) or without information on the clinical process (*n* = 2), were excluded from the study.

### 4.2. Data Collection

Clinical data included age, gender, family history of CRC, clinical presentation, tumour site, and CEA levels. Anatomopathological data included macroscopic aspect, tumour size, histological type, staging, tumour differentiation and presence of lymphatic or vascular invasion. Postoperative data included length of hospital stay, hospital readmission, postoperative complications, Clavien–Dindo classification (CDC) of postoperative morbidity severity and 30-day mortality. The follow up data embraced overall and recurrence-free survival rates.

### 4.3. Image Analysis

SMI was measured, on preoperative abdominal CT, using the National Institutes of Health (NIH) ImageJ®semi-automatic software. The cross-sectional images, at L3 level, were anonymised and transferred in Digital Imaging and Communications in Medicine (DICOM) format, with the collaboration of a radiologist, and opened using the ImageJ®software. Total abdominal muscle area was assessed automatically, after the manual outlining of the outer and inner skeletal muscle boundaries. The distinction between different tissues is based on Hounsfield Units (HU). A threshold range of $-29$ to $+150$ HU was used for skeletal muscle [22,23]. This measure was evaluated by one investigator and then normalised for patient stature ($cm^2/m^2$) (SMI = total abdominal muscle area/height$^2$) (Figure 3).

Sarcopenia was defined using gender-specific cut-off points of 38.4 $cm^2/m^2$ for female patients and 52.4 $cm^2/m^2$ for the male patients, based on the method of Prado et al. [24].

The research protocol was submitted to Subcommittee on Ethics for Life and Health Sciences of Minho University (CECVS 054/2017) and the Ethics Committee for Health of Hospital de Braga (CESHB 79/2017), and received approval from both committees.

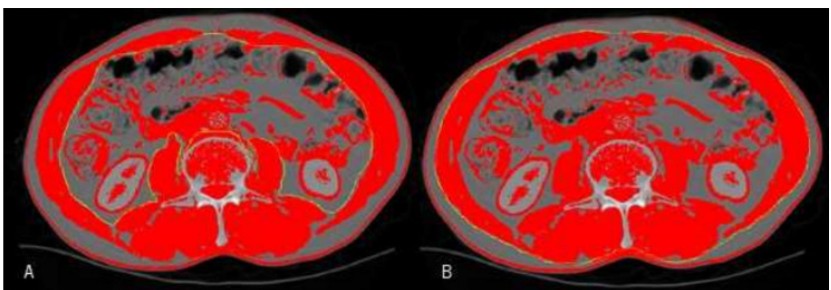

**Figure 3.** Cross-sectional abdominal computed tomography (CT), at L3 level, with the manual outline of the inner (Figure 1A) and external (Figure 1B) boundaries (yellow line) of muscle mass and the application of the threshold from −29 to +150 UH (red). Multiple muscles were identified for the measurement, including psoas muscle, erector spinae, quadratus lumborum, transversus abdominis, external and internal obliques and rectus abdominis.

*4.4. Statistical Analysis*

All statistical analyses were performed using the Statistical Package for the Social Science (SPSS®), version 23.0 for Windows®.

For all variables, a descriptive analysis was performed, expressed by median (*Mdn*) and interquartile range (*IQR*), for the quantitative data, and absolute (*n*) and relative frequencies (%) for the qualitative data.

To assess associations between sarcopenia and qualitative data, Pearson Chi-Square Test ($\chi^2$) or the Fisher's Exact Test were performed, as appropriate. For quantitative data, the differences between groups were compared by the Mann–Whitney test.

A univariate and multivariate logistic regression was performed to assess whether sarcopenia was an independent predictor of severe morbidity, Odds Ratios (*OR*) at 95% confidence intervals (*CI*) were presented. In order to make the model as parsimonious as possible, the predictors that were included in the univariate analyses were selected according to the literature, and if they were statistically significant predictors in the univariate analyses or presented a level of significance lower than 0.20, they were included in the multivariate analysis.

The overall survival and recurrence-free survival were analyzed by the Kaplan–Meier method and the Log-rank test. A univariate and multivariate Cox regression, with the same principle of data choice of variables as described above, were performed to assess whether sarcopenia was an independent predictor of recurrence-free survival, Hazard Ratios (HR) at 95% CI were presented.

All tests were considered significant for a *p*-value less than 0.05.

## 5. Conclusions

The presence of sarcopenia, assessed by muscle mass, at L3 level, adjusted for stature, was a strong predictor of major postoperative morbidity, taking place in more than three-quarters of the sarcopenic patients, when a postoperative complication happened. Some complications are related to the surgery itself, such as anastomosis leakage or intraabdominal abscess; however, sarcopenia remained as an independent predictor of major morbidity in a multivariate analysis. No associations were found between sarcopenia and a longer hospital stay, 30-day mortality and lower global and recurrence-free survival.

Thus, this study showed that the measurement of muscle mass in preoperative CT, in patients with CRC who undergo surgical resection, is a simple and inexpensive diagnostic method of sarcopenia, since it is done for CRC staging. In this way, it is possible to predict, in the presence of this physical condition, a greater probability of worse surgical outcomes and, by this way, to act in a timely and preventive manner.

**Author Contributions:** Conceptualization, C.C. and S.F.M.; methodology, A.P., P.S., C.C. and S.F.M.; software, C.C.; validation, C.C. and S.F.M.; formal analysis, M.P. and S.F.M.; investigation, M.P., A.P. and P.S.; data curation, M.P.; writing—original draft preparation, M.P.; writing—review and editing, A.P., P.S., C.C. and S.F.M.; supervision, S.F.M.; project administration, S.F.M. All authors have read and agreed to the published version of the manuscript.

**Funding:** This research received no external funding.

**Conflicts of Interest:** The authors declare no conflict of interest.

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
