# Peer review of "Sarcopenia as a Risk Factor of Morbimortality in Colorectal Cancer Surgery"

_gastrointestdisord, doi:10.3390/gidisord2020010_

Round 1
Reviewer 1 Report
The paper is of interest.
I think the imaging parts deserves more consideration. Do you think it may be helpful and reproducibile in clinics? It is a validated tool in oncology? Are there any other series reported on this outcome?
Why did you choose only CRC? Can you please compare your results on CRC with other diseases of interest? I think gastric cancer it is probably the major pathology of interest concerning the topic of sarcopenia.
Some examples:
J Cancer Res Clin Oncol. 2019 Sep;145(9):2365-2373. doi: 10.1007/s00432-019-02971-7. Epub 2019 Jul 6. Liver Int. 2020 Mar;40(3):704-711. doi: 10.1111/liv.14314. Epub 2019 Dec 11Author Response
- I think the imaging parts deserves more consideration. Do you think it may be helpful and reproducibile in clinics? It is a validated tool in oncology? Are there any other series reported on this outcome?
Response: We thank the reviewer for the comment. This assessment can be very easily implemented in the clinic. CT is performed for CRC staging and as we can see, even though this work having been performed retrospectively, it was possible, in most of the exams, for the imagiologist to evaluate what was intended in terms of image. If this evaluation starts to be carried out prospectively, in which the images are captured in order to also make these measurements, it will be possible to optimize this data and generalize it to all cancer patients. The previous knowledge of sarcopenic patients and the association of this factor with greater morbidity or side effects of chemotherapy will contribute in the future to a better individualization of therapeutics
- Why did you choose only CRC? Can you please compare your results on CRC with other diseases of interest? I think gastric cancer it is probably the major pathology of interest concerning the topic of sarcopenia.
Response: We thank the reviewer for the comment. We chose the CCR because we are part of a Colorectal Unit, in which this cancer is the bulk of our treatment. Hence the focus, since we wanted to know whether sarcopenia influenced our results or not.
As the topic has become interesting for the rest of the Surgery department, we are at this stage extending the study to other units, namely the esophagogastric unit and the hepatobiliary unit, but still with no publishable results.
Reviewer 2 Report
Thank you for the interesting manuscript. It is clearly structured and presents the data well. The language could be improved in some passages.
Prior to publication some points should be addressed:
- Clavien-Dindo classification is missing in a lot of patients (only available in 92/272). There is risk of bias. Please comment on these missing values as this concerns one of the main findings of the study.
Minor points:
- Consider adding the p-values to the figures. Otherwise, the reader might miss that the difference of the curves is not statistically significant, if looking only at the figures.
- Table 6: "Valuables" in the heading is the incorrect english word
- Table 6: It should be "Adjusted HR" (not "Unadjusted") in the multivariate column.
Author Response
- Clavien-Dindo classification is missing in a lot of patients (only available in 92/272). There is risk of bias. Please comment on these missing values as this concerns one of the main findings of the study.
Response: We thank the reviewer for the comment. As we can see in table 3; 92 patients correspond to the patients who developed morbidity in a total of 272, the reminder did not show morbidity, therefore the classification Clavien Dindo is not missing in this patients, the classification is not applicable to these patients (only to the 92 who developed morbility)
Minor points:
- Consider adding the p-values to the figures. Otherwise, the reader might miss that the difference of the curves is not statistically significant, if looking only at the figures.
Response: We thank the reviewer for the comment. P-values were added to the figures
- Table 6: "Valuables" in the heading is the incorrect english word
Response: We thank the reviewer for the comment. The word “valuables” was removed from the table.
- Table 6: It should be "Adjusted HR" (not "Unadjusted") in the multivariate column.
Response: We thank the reviewer for the comment. The alteration has been made